# Cross-informant ratings on emotional and behavioral problems in Nepali adolescents: A comparison of adolescents' self-reports with parents' and teachers' reports

Sirjana Adhikari [1,2]*, Jasmine Ma[2], Suraj Shakya[3], Per Håkan Brøndbo[1], Bjørn Helge Handegård[4], Anne Cecilie Javo[5]

1 Department of Psychology, Faculty of Health Sciences, UiT, The Arctic University of Norway, Tromsø, Norway, 2 CWIN-Nepal, Ravi Bhawan, Kathmandu, Nepal, 3 Department of Psychiatry and Mental Health, Institute of Medicine, Tribhuvan University, Kathmandu, Nepal, 4 Regional Centre for Child and Youth Mental Health and Child Welfare -North, Faculty of Health Sciences, UiT, The Arctic University of Norway, Tromsø, Norway, 5 Sami National Competence Center for Mental Health (SANKS), Sami Klinihkka, Finnmark Hospital Trust, Karasjok, Norway

* adhikarisirjana16@gmail.com

**Data Availability Statement:** All relevant data are within the Supporting Information file, names as S1 Dataset.

## Abstract

### Background

Studies on cross-informant agreement on adolescents' emotional and behavioral problems (EBPs) are sparse in low- and middle-income countries. This study aimed to assess parent-adolescent and teacher-adolescent agreement on EBPs and associated factors in Nepal.

### Methods

This cross-sectional survey included 1904 school-going adolescents aged 11–18, enrolled in government and private schools located in sixteen districts of Nepal. The Nepali versions of the Youth Self Report, Child Behavior Checklist, and Teacher's Report Form were administered to assess EBPs reported by adolescents, their parents, and teachers, respectively. Repeated measures analysis of variance (ANOVA) was done to assess mean differences in problem scores. Pearson's correlation was used to assess cross-informant agreement. Linear regression analysis was used to explore factors associated with cross-informant discrepancies in EBPs.

### Results

Adolescents reported significantly more problems than their parents and teachers. Mean Total Problem scores for the 90 common items in the adolescents' self-reports, parent reports, and teacher reports were 34.5 (standard deviation [SD]: 21.4), 24.1 (SD = 19.2), and 20.2 (SD = 17.5) respectively. Parent-adolescent agreement on Total Problems was moderate, whereas teacher-adolescent agreement was low. The parent-adolescent agreement was moderate to low for the two broadband scales and all syndrome scales, whereas the teacher-adolescent agreement was low for all scales. Female gender and ethnic

**Funding:** This study was funded by the Norwegian Partnership Program for Global Academic Cooperation (NORPART), 2018/10039 project: "Collaboration in Higher Education in Mental Health between Nepal and Norway", and the Child Workers in Nepal (CWIN-Nepal). The expenditures of the research work were funded by the NORPART project, and the salary of the principal investigator was funded by the CWIN-Nepal. URL: 1. NORPART: https://diku.no/en/programmes/norpart-norwegian-partnership-programme-for-global-academic-cooperation/. 2. CWIN-Nepal: https://www.cwin.org.np/. The charges for online publication have been funded by a grant from the publication fund of UiT-The Arctic University of Norway. The funders had no role in study design, data collection and analysis, decision to publish, or preparation of the manuscript.

**Competing interests:** The authors have declared that no competing interests exist.

minority status impacted both parent-adolescent and teacher-adolescent discrepancies. Family stress/conflicts impacted parent-adolescent discrepancies, while academic performance impacted teacher-adolescent discrepancies.

## Conclusions

Nepali adolescents reported more EBPs than their parents and teachers. The agreement between adolescents' self-reports and reports by their parents and teachers was moderate to low. Gender, caste/ethnicity, family stress/conflicts, and academic performance were associated with cross-informant discrepancies. It is crucial to collect information from different sources, consider context-specific needs, and discern factors influencing cross-informant discrepancies to accurately assess adolescents' EBPs and develop personalized approaches to treatment planning.

## Introduction

Adolescents may display emotional and behavioral problems (EBPs) in some contexts and not in others (e.g., home versus school). Information from different informants helps to identify important consistencies and differences in adolescents' functioning across diverse settings, and as seen from different perspectives. Such information is important to assist mental health professionals in accurately assessing adolescent problem behaviors and in tailoring specific treatments for those in need [1–3]. Therefore, an established gold standard in the assessment of mental health in adolescents is to combine information from various sources, including adolescents, parents, and teachers [3–6]. Adult informants are the most common sources and can provide critical information about how adolescents function in everyday settings such as the home and school [2, 7, 8]. However, for a comprehensive evaluation of mental health, it is also vital to draw information from the adolescents themselves [9, 10]. Indeed, parents and teachers may tend to minimize adolescents' problems, and thus, adolescents may be less likely to receive early diagnosis and proper treatment if they are not questioned directly. Further, discrepancies and changes in discrepancies over time of parent- and self-reports, and of teacher- and self-reports of internalizing and externalizing problems may contribute to the prediction of adult internalizing and externalizing disorders [11].

According to the literature, cross-informant agreement on adolescent EBPs is typically only low to moderate [4, 12–14], a finding that is robust across countries. Rescorla and colleagues explored whether the tendencies of adolescents, parents, and teachers to endorse different items could limit the cross-informant agreement. They demonstrated that, when averaged across all societies, the mean $Q$ correlation between ratings was high, indicating that different informants tended to give low, medium, or high ratings to the same items [14]. However, discrepancies in the information obtained from different informants could occur for many reasons, such as differences in the way adolescents behave in different contexts, differences in the informant's relationship with the child, and cultural differences in the perception of and thresholds for reporting a child's behavior [15]. Further, studies have demonstrated that several other factors have an impact on the level of agreement, such as adolescents' age and gender [4, 11, 16, 17], ethnicity [18], the experience of serious life events [16], parenting [12], parental hostility and family status [17], family conflict and dysfunction [19], and parents' school engagement and contact with the teacher [12].

## Adolescent self-reports compared to parent reports on emotional and behavioral problems

An earlier meta-analysis included studies from twenty-five different societies that used the Youth Self-Report (YSR) to explore self-reported EBPs among adolescents. It showed that adolescents tended to report more problems in all problem domains than their parents [14]. A recent meta-study based on raw scores of EBPs confirmed this finding [4]. Further, a recent study from Japan used the Strengths and Difficulties Questionnaire (SDQ) and found that adolescent reports yielded higher total difficulty scores and higher scores on all SDQ subscales than corresponding parent reports [10]. Another recent study from the Netherlands reported comparable results [20]. Moreover, a study from Taiwan found that adolescents reported higher symptom scores than their parents [8].

Studies suggest that adolescents report higher internalizing problems than parents, such as somatization and anxiety [7, 12]. The lower parent reports of internalizing problems suggest that parents tend to underestimate adolescents' emotional distress [21].

Meta-analyses and studies on parent-adolescent agreement on EBPs across many countries have indicated small or moderate agreement [4, 13, 14, 22, 23]. However, parent-adolescent agreement in Western countries has long been the focus of extensive research, whereas parent-adolescent agreement in less-examined low-middle-income countries and ethnic minority populations has received less attention. Hence, we do not know whether the same pattern applies to all cultures. For instance, a recent study by Sinclair and colleagues which examined parent-adolescent agreement on EBPs in indigenous and non-indigenous dyads, found high agreement in the indigenous group, but low to moderate agreement in the non-indigenous group, suggesting possible cultural differences [18]. In contrast to most studies from Western countries, studies from Asian countries have suggested moderate to high parent-adolescent agreement on EBPs, with moderate agreement ($r = 0.4$) in a recent study from Taiwan [8] and high agreement ($r = 0.6$) in a study from China [16]. A study from the South Asian country of Sri Lanka suggested moderate agreement between parents and adolescents [24]. However, more studies from less-examined Asian countries, such as Nepal, are warranted to explore further possible cultural differences in cross-informant correlations in Asian countries.

## Adolescent self-reports compared to teacher reports on emotional and behavioral problems

Few studies have compared mean EBPs between teacher reports and adolescent reports. Teachers (like parents) report fewer total problems than adolescents [4, 7, 25, 26]. To date, studies have shown that, whereas teachers seem to report considerably fewer internalizing problems than adolescents [4, 7, 27, 28], they seem to report more externalizing problems, which might be connected to the fact that externalizing problems often disrupt the school environment and therefore tend to be more easily identified by teachers [4, 27].

Further, few studies have been performed on teacher-adolescent agreement on EBPs, i.e., for the same child [4]. In most studies, teacher-adolescent agreement on EBPs was low to moderate [4, 25, 29]. Further, cross-informant agreement studies have found lower concordance on EBPs between teachers and adolescents than between parents and adolescents [7, 30]. Additionally, the teacher-adolescent agreement was higher for Externalizing Problems than for Internalizing Problems and Total Problems [30].

In conclusion, the international literature suggests that there is a notable lack of consensus among parents, teachers, and adolescents regarding adolescents' EBPs. However, a search of the databases (PubMed and PsycINFO) revealed a scarcity of research on cross-informant agreements in South Asian countries. In a study from Sri Lanka, the interrater agreement

between teacher reports and adolescent self-reports was low [24]. A recent Nepali study on parent-teacher agreement in a wider age group (6–18 years) reported moderate to low agreement [31], but so far, no study has compared the agreement between adolescents and their parents and teachers in Nepal. Further research is needed in less investigated low- and middle-income countries (LMICs) like Nepal to determine if the observations regarding cross-informant discrepancies remain consistent in these regions. Also, the impact of cultural and societal factors on cross-informant agreement remains underexplored. Identifying factors that may contribute to the contextual variability in reporting symptoms can enhance our understanding of the nature of adolescents' EBPs. Moreover, it is essential to understand the sources and nature of discrepancies to formulate effective intervention strategies and to develop culturally sensitive assessments of adolescent EBPs.

The present study aims to fill the aforementioned knowledge gap for Nepal by (1) examining differences in mean scores of EBPs between parents and adolescents and between teachers and adolescents, (2) examining cross-informant correlations between parents and adolescents, and between teachers and adolescents, and (3) examining the impact of adolescents' age, gender, caste/ethnicity, the experience of serious life events/trauma, family stress/conflicts, and academic performance on cross-informant discrepancies.

## Materials and methods

This study is part of a larger, cross-sectional population-based survey on EBPs in the general child population of Nepal (aged 6–18 years) [31–33]. A more detailed description of the procedure and background instrument has been reported in a previous study [32].

### Participants and procedure

The present study included 1904 adolescents who were attending school during the time of data collection, in addition to their mothers and schoolteachers. Based on the population distribution of the three main ecological/geographical regions of Nepal, sixteen districts were selected (three from the Mountains region, six from the Middle Hills region, six from the Terai region, and the Kathmandu district). Four schools (two governmental and two private) were selected from each district. Both districts and schools were selected using a purposive sampling method based on accessibility and feasibility. The study was countrywide and required an extensive amount of time and money to accomplish. Hence, purposive sampling was chosen for cost-effectiveness and ease of data collection and travel. However, the students, i.e., three boys and three girls aged 11–18 years from each grade level (grades 6–10), were randomly selected using random number tables, regardless of caste/ethnicity or other background variables. The overall participation rate was 99.2% for the adolescent sample and parent sample, and 99.1% for the teacher sample.

A team of trained research assistants performed the original data collection work in the different districts, monitored by the project leader [32]. A meeting with the school management was conducted at each school. After approval, an invitation letter was sent to the parents to inform them about the project. Each adolescent completed the YSR in their school setting. Parents of the selected adolescents completed the Child Behavior Checklist (CBCL) to report the adolescent's problem behaviors. Teachers of the selected adolescents completed the Teacher's Report Form (TRF). Parents were also asked to complete a background questionnaire. Research assistants helped illiterate parents in completing the questionnaires. Data were collected from September 2017 to January 2018. Plotting of data was done manually during the first half of 2018 by research assistants [32]

### Ethics statement

Before commencing the study, ethical approval was obtained from the Ethical Review Board of the Nepal Health Research Council (NHRC), (reference number: 1875; registration number: 71/2017). Written informed consent was obtained from the parents of all selected adolescents following the rules of the NHRC. Both verbal and written information about the study was provided to the teachers, parents, and adolescents. The records from the study were kept strictly confidential and locked down.

### Measures

We used the Nepali versions of the Youth Self Report (YSR/11-18), the Child Behavior Checklist (CBCL/11-18), and the Teacher's Report Form (TRF/11-18), which had been translated into Nepali in connection with a previous Nepali Ph.D. dissertation [34]. All these instruments are part of the Achenbach System of Empirically Based Assessment (ASEBA), which contains instruments that collect data on children's and adolescents' problems [35]. ASEBA instruments are empirically based, are widely used to assess adaptive and maladaptive functioning in children and adolescents, and have been translated into numerous languages [36]. Confirmatory factor analyses have supported their syndrome structures in dozens of societies [37].

### Youth self report, child behavior checklist, and teacher's report form

The YSR was used to assess self-reported EBPs in adolescents. It targets those aged 11–18 years and consists of 105 problem items, including a broad range of problem behaviors [35]. It was modeled after the CBCL and has a similar format, except that the items are worded in the first person. The YSR has counterparts of 105 of the CBCL/6-18 problem items and counterparts of 93 TRF problem items [35]. The YSR was found to be suitable for use in an earlier Nepali study, with good internal consistency [38]. Our recently published paper on self-reported EBPs among Nepali adolescents also documented acceptable to good psychometric properties of the YSR, based on the same study sample [9]. When completing the questionnaire, adolescents indicate the extent to which each item describes their behavior over the last 6 months by selecting a score of 0 (not true), 1 (somewhat or sometimes true), or 2 (very true or often true) for each item.

The CBCL is designed for parents or caregivers. The TRF is designed for schoolteachers to report on competencies (including academic performance) and EPBs in children aged 6–18 years. In the present study, we used the academic performance item and the problem items for the age group 11–18 years only. The CBCL and TRF have 120 problem items including two open-ended questions about the child's problems. As in the YSR, all items are rated on a 3-point Likert scale as 0 (not true), 1 (somewhat or sometimes true), or 2 (very true or often true). CBCL ratings are based on the child's functioning over the last 6 months, whereas TRF ratings are based on the child's functioning over the last 2 months. Overall good internal consistency has been reported for the CBCL/6-18 years (Cronbach's alpha = 0.71 to 0.88 for the eight syndrome scales) and the TRF/6-18 years (Cronbach's alpha = 0.74 to 0.89 for the eight syndrome scales) in previous Nepali studies [31, 32]. Furthermore, acceptable to good internal consistency was observed for the syndrome scales of the YSR, CBCL, and TRF within the present adolescent sample (Table 1).

The YSR, CBCL, and TRF have ninety common items, as well as other different items appropriate to the home or school context, respectively. In all these instruments, problems are scored on a Total Problems scale, two broadband scales (Internalizing and Externalizing Problems), and eight syndrome scales. The syndrome scales of Anxious/Depressed, Withdrawn/Depressed, and Somatic Complaints are combined to form the Internalizing scale, whereas

**Table 1. Internal consistencies of the Youth Self Report (YSR), Child Behavior Checklist (CBCL), and Teacher's Report Form (TRF) for Nepali adolescents (N = 1904).**

| Syndrome scales | Cronbach's alpha α | | |
|---|---|---|---|
| | YSR* | CBCL | TRF |
| Anxious/Depressed | 0.76 | 0.74 | 0.79 |
| Withdrawn/Depressed | 0.68 | 0.69 | 0.78 |
| Somatic Complaints | 0.78 | 0.78 | 0.76 |
| Social Problems | 0.69 | 0.69 | 0.75 |
| Thought Problems | 0.73 | 0.72 | 0.72 |
| Attention Problems | 0.73 | 0.79 | 0.91 |
| Rule-Breaking Behavior | 0.71 | 0.73 | 0.74 |
| Aggressive Behavior | 0.84 | 0.86 | 0.88 |

*Reprinted from: Adhikari S, Ma J, Shakya S, Brøndbo PH, Handegård BH, Javo AC. Self-reported emotional and behavioral problems among school-going adolescents in Nepal-A cross-sectional study. PLOS ONE. 2023 Jun 23;18 (6):e0287305. doi: 10.1371/journal.pone.0287305.

Rule-Breaking Behavior and Aggressive Behavior are combined to form the Externalizing scale. The Social Problems, Attention Problems, and Thought Problems scales do not belong to either of the broadband scales but are included in the Total Problems scale.

## Background questionnaire

Mothers completed a background questionnaire that collected information on variables like age, gender, caste/ethnicity, the experience of serious life events/trauma affecting the child, and family stress/conflicts. Caste/ethnicity was assessed by presenting the list of caste groups according to the Nepal census [39] and parents were asked to put a tick mark on the relevant option. Responses were grouped into four categories (Hindu high caste groups, Indigenous groups/ethnic minorities, Hindu low caste groups, and Others). The question on serious life events/trauma was presented as: "Has the child experienced any serious life events or trauma during the past 12 months that might have affected him/her psychologically?", and the options given were "yes" or "no". Family stress/conflicts were assessed with the question: "Have there been any conflicts between family members causing stress in the family during the past 6 months?" Response options were high, moderate, or low levels of stress.

## Statistical analyses

Statistical Package for the Social Sciences (SPSS) statistics version 26.0 for Windows was used for all analyses. To compare the mean scores of the ninety common items in the YSR, CBCL, and TRF, repeated measures ANOVA was used. The Pearson correlation coefficient (r) was used to assess parent-adolescent and teacher-adolescent agreement based on the correlations between the YSR and the CBCL scale scores and between the YSR and the TRF scale scores. Cohen (1988) suggests that r = 0.10 indicates a typical small effect, 0.30 indicates a typical medium effect, and 0.50 indicates a typical large effect [40]. Multiple regression analysis was used to explore factors that influenced parent-adolescent and teacher-adolescent discrepancies. Effect sizes for individual independent variables in the regression analysis were computed using partial eta squared (partial $\eta^2$). A partial $\eta^2 = 0.01$ is a typical small effect, 0.06 is a typical medium effect, and 0.14 is a typical large effect [40]. The significance level used for all tests was 0.01.

# Results

## Adolescent self-reports compared to parent reports on emotional and behavioral problems

The mean scores of the ninety common items on the CBCL and YSR for Total Problems, Internalizing Problems, Externalizing Problems, and the different syndrome scales were higher in the adolescent self-reports than in parent reports. We found significant differences between the adolescent and parent reports for all problem scales, and the effect size was large for Total Problems, Internalizing Problems, Anxious/Depressed, Withdrawn/Depressed, Social Problems, and Thought Problems (partial $\eta^2$ between 0.15 and 0.20). The effect size was small to medium for Externalizing Problems, Attention Problems, Aggressive Behaviors, and Rule-Breaking Behaviors (partial $\eta^2$ between 0.03 and 0.06) (Table 2).

## Adolescent self-reports compared to teacher reports on emotional and behavioral problems

The mean scores of the ninety common items on the TRF and YSR for Total Problems, Internalizing Problems, Externalizing Problems, and the different syndrome scales were higher in adolescent self-reports than in teacher reports. Consistent with parent-adolescent comparisons, there were significant differences between the teacher and adolescent reports on all problem scales. The effect size was large ($\eta^2 > 0.1$) for all scales except Attention Problems and Rule-Breaking Behavior (Table 3).

## Parent-adolescent and teacher-adolescent agreement on emotional and behavioral problems

Pearson's correlations showed moderate parents-adolescent agreement on Total Problems ($r = 0.31$, $p < 0.01$), Internalizing Problems ($r = 0.30$, $p < 0.01$), and Externalizing Problems ($r = 0.32$, $p < 0.01$). Similarly, there was moderate agreement on Somatic Complaints, Social Problems, Aggressive Behaviors, and Rule-Breaking Behaviors. We found a low agreement for the other syndrome scales (Anxious/Depressed, Withdrawn/Depressed, Attention Problems, and Thought Problems).

**Table 2. Comparison between the magnitude of adolescent-reported and parent-reported EBPs (N = 1904).**

| Problem scales | CBCL$_{90}$ | YSR$_{90}$ | F | Partial eta squared |
| --- | --- | --- | --- | --- |
| | Mean (SD) | Mean (SD) | | |
| Total Problems | 24.10 (19.12) | 34.51 (21.37) | 356.72* | 0.16 |
| Internalizing problems | 9.14 (7.45) | 13.63 (8.44) | 426.38* | 0.18 |
| Externalizing Problems | 6.22 (6.09) | 7.99 (6.40) | 111.09* | 0.06 |
| Anxious/Depressed | 4.16 (3.51) | 6.10 (3.96) | 340.97* | 0.15 |
| Withdrawn/Depressed | 2.42 (2.47) | 3.95 (2.75) | 407.58* | 0.18 |
| Somatic Complaints | 2.56 (2.86) | 3.59 (3.11) | 165.64* | 0.08 |
| Social Problems | 3.23 (2.93) | 4.87 (3.22) | 354.01* | 0.16 |
| Attention Problems | 3.60 (3.26) | 4.24 (3.15) | 50.39* | 0.03 |
| Thought Problems | 1.43 (2.03) | 2.99 (2.81) | 481.41* | 0.20 |
| Aggressive Behaviors | 4.36 (4.45) | 5.44 (4.59) | 78.76* | 0.04 |
| Rule-Breaking Behaviors | 1.86 (2.14) | 2.55 (2.39) | 119.82* | 0.06 |

*p < 0.01; CBCL: Child Behavior Checklist, YSR: Youth Self Report, SD: standard deviation, CBCL$_{90}$, and YSR$_{90}$: mean scores for the 90 common items.

**Table 3. Comparison between the magnitude of adolescent-reported and teacher-reported EBPs (N = 1904).**

| Problem scales | TRF$_{90}$ | YSR$_{90}$ | F | Partial eta squared |
|---|---|---|---|---|
| | Mean (SD) | Mean (SD) | | |
| Total Problem | 20.15 (17.46) | 34.51 (21.37) | 675.58* | 0.26 |
| Internalizing problems | 7.27 (6.61) | 13.63(8.44) | 857.60* | 0.31 |
| Externalizing Problems | 5.31 (6.07) | 7.99 (6.40) | 231.76* | 0.11 |
| Anxious/Depressed | 3.35 (3.12) | 6.10 (3.96) | 690.23* | 0.27 |
| Withdrawn/Depressed | 2.51 (2.63) | 3.95 (2.75) | 317.03* | 0.14 |
| Somatic Complaints | 1.41 (2.12) | 3.59 (3.11) | 819.47* | 0.31 |
| Social Problems | 2.46 (2.70) | 4.87 (3.22) | 805.76* | 0.31 |
| Attention Problems | 3.50 (3.13) | 4.24 (3.15) | 70.66* | 0.04 |
| Thought Problems | 1.21 (1.87) | 2.99 (2.80) | 601.25* | 0.24 |
| Aggressive Behaviors | 3.37 (4.09) | 5.44 (4.59) | 282.30* | 0.13 |
| Rule-Breaking Behaviors | 1.94 (2.41) | 2.55 (2.39) | 76.86* | 0.04 |

*$p < 0.01$; TRF: Teacher's Report Form, YSR: Youth Self Report, SD: standard deviation; TRF$_{90}$ and YSR$_{90}$: mean scores for the 90 common items.

In contrast to the parent-adolescent agreement, teacher-adolescent agreement on EBPs was low for Total Problems, Internalizing Problems, Externalizing Problems, and all syndrome scales. The teacher-adolescent agreements ranged from moderate (r = 0.26) for Externalizing Problems to low (r = 0.13) for Thought Problems (Table 4).

## Factors associated with parent-adolescent discrepancies

We used multiple regression analysis to examine the influence of background variables such as adolescents' age, gender, caste/ethnicity, the experience of negative/traumatic live events, and family stress/conflicts. To measure parent-adolescent discrepancies, we subtracted YSR scores from CBCL scores for Total, Internalizing, and Externalizing Problems. For Total Problems, the mean parent-adolescent discrepancy score was −10.36 (SD = 23.86), for Internalizing Problems it was −4.48 (SD = 9.43), and for Externalizing Problems it was −1.76 (SD = 7.28). We entered all the background variables in the model to explore their association with parent-adolescent discrepancies and to examine the main effects of these background variables.

**Table 4. Parent-adolescent and teacher-adolescent agreement on EBPs in Nepali adolescents (N = 1904).**

| Problem scales | CBCL-YSR | TRF-YSR |
|---|---|---|
| | Pearson's Correlation (*r*) | Pearson's Correlation (*r*) |
| Total Problems | 0.31* | 0.25* |
| Internalizing Problems | 0.30* | 0.23* |
| Externalizing Problems | 0.32* | 0.26* |
| Anxious/Depressed | 0.26* | 0.20* |
| Withdrawn/Depressed | 0.21* | 0.15* |
| Somatic Complaints | 0.33* | 0.24* |
| Social Problems | 0.33* | 0.23* |
| Attention Problems | 0.27* | 0.25* |
| Thought Problems | 0.23* | 0.13* |
| Aggressive Behaviors | 0.31* | 0.25* |
| Rule-Breaking Behaviors | 0.30* | 0.23* |

*$p < 0.01$; CBCL: Child Behavior Checklist, YSR: Youth Self Report, TRF: Teacher's Report Form.

For Total Problems, 2.4% of the variance ($R^2 = 0.024$) in parent-adolescent discrepancies was explained by the background variables. Caste/ethnicity was significantly associated with the discrepancy score. The parent-adolescent discrepancy was larger in the indigenous groups/ethnic minorities than in the Hindu high-caste groups. Family stress/conflicts also impacted the discrepancy. Our estimates showed that in families with a high level of stress/conflict, the parents gave higher scores for EBPs than adolescents.

Background variables accounted for 2.6% of the variability in parent-adolescent discrepancies regarding Internalizing Problems. Gender, caste/ethnicity, and family stress/conflicts were significantly associated with the discrepancy scores. Parent-adolescent discrepancies were larger for female adolescents, indigenous groups/ethnic minorities, and families with a high level of stress/conflict.

For Externalizing Problems, the background variables accounted for only 1.8% of the variance in parent-adolescent discrepancies. Caste/ethnicity was the only variable that was significantly associated with parent-adolescent discrepancies. Here, too, parent-adolescent discrepancy scores were significantly larger in indigenous groups/ethnic minorities than in the Hindu high-caste groups. However, the effect sizes were small ($\eta^2 \leq 0.01$) for all situations (Table 5).

## Factors associated with teacher-adolescent discrepancies

Again, we used multiple regression analyses to examine the influence of background variables like adolescents' age, gender, caste/ethnicity, the experience of serious life events/trauma, family stress/conflicts, and academic performance. To measure teacher-adolescent discrepancies, we subtracted YSR scores from TRF scores for Total, Internalizing, and Externalizing Problems. Mean scores for teacher-adolescent discrepancies were −14.36 (SD = 24.02) for Total Problems, −6.36 (SD = 9.44) for Internalizing Problems, and −2.67 (SD = 7.65) for Externalizing Problems. To explore the association between the background variables and teacher-adolescent discrepancies in EBPs, we entered all variables in the model to examine the main effects of the discrepancies. For Total Problems, 2.6% of the variance in the discrepancy score was explained by the background variables. Among the different variables, adolescents' gender, caste/ethnicity, and academic performance were all significantly associated with the discrepancy score.

**Table 5. Factors associated with parent-adolescent discrepancies.**

| Background variables | CBCL-YSR discrepancies | | | | | | | | |
|---|---|---|---|---|---|---|---|---|---|
| | **Total Problems** | | | **Internalizing Problems** | | | **Externalizing Problems** | | |
| | **B** | **F** | **Partial η²** | **B** | **F** | **Partial η²** | **B** | **F** | **Partial η²** |
| **Age** | −0.29 | 0.86 | 0.00 | 0.09 | 0.52 | 0.00 | −0.24 | 6.29 | 0.00 |
| **Gender** (Reference group = Male) | −2.30 | 4.41 | 0.00 | −1.43 | 10.87** | 0.01 | −0.25 | 0.54 | 0.00 |
| **Caste/ethnicity** (Reference group = Hindu high castes group) | | 10.42** | 0.01 | | 8.99** | 0.01 | | 6.86** | 0.01 |
| Indigenous groups/ethnic minorities | −5.24 | 20.07** | 0.01 | −1.89 | 16.72** | 0.01 | −1.28 | 12.82** | 0.01 |
| Hindu low-caste groups | −0.43 | 0.04 | 0.00 | 0.06 | −0.01 | 0.00 | 0.05 | 0.01 | 0.00 |
| **Negative/traumatic life events in the past 12 months** (Reference group = No) | 4.75 | 5.05 | 0.00 | 2.03 | 5.87 | 0.00 | 1.23 | 3.57 | 0.00 |
| **Family stress/conflicts** (Reference group = low level of stress) | | 5.13* | 0.01 | | 5.08* | 0.01 | | 3.08 | 0.00 |
| Moderate level of stress/conflicts | 1.96 | 1.08 | 0.00 | 0.90 | 1.46 | 0.00 | 0.91 | 2.51 | 0.00 |
| High level of stress/conflicts | 11.81 | 9.49 * | 0.01 | 4.59 | 9.06* | 0.01 | 2.35 | 3.96 | 0.00 |
| | $R^2 = 0.024$ | | | $R^2 = 0.026$ | | | $R^2 = 0.018$ | | |

*p < 0.01

**p < 0.001, B = Unstandardized Regression Coefficient, F = F statistics, $\eta^2$: Eta squared. CBCL: Child Behavior Checklist, YSR: Youth Self Report.

**Table 6. Factors associated with teacher-adolescent discrepancies in Nepali adolescents (N = 1904).**

| Background variables | TRF-YSR discrepancies | | | | | | | | |
|---|---|---|---|---|---|---|---|---|---|
| | Total Problems | | | Internalizing Problems | | | Externalizing Problems | | |
| | B | F | Partial $\eta^2$ | B | F | Partial $\eta^2$ | B | F | Partial $\eta^2$ |
| **Age** | −0.27 | 0.77 | 0.00 | −0.08 | 0.39 | 0.00 | −0.07 | 0.48 | 0.00 |
| **Gender** (Reference group = Male) | −4.02 | 13.29** | 0.01 | −1.30 | 8.95* | 0.01 | −1.33 | 14.28** | 0.01 |
| **Caste/ethnicity** (Reference group = Hindu high caste groups) | | 13.61** | 0.02 | | 10.41** | 0.01 | | 12.28** | 0.01 |
| Indigenous groups/ethnic minorities | −6.07 | 26.42** | 0.01 | −2.09 | 20.16** | 0.01 | −1.80 | 22.85** | 0.01 |
| Hindu low-caste groups | −0.74 | 0.14 | 0.00 | −0.25 | 0.10 | 0.00 | 0.06 | 0.00 | 0.00 |
| **Negative/traumatic life events in the past 12 months** (Reference group = No) | −3.39 | 2.56 | 0.00 | 0.85 | 1.02 | 0.00 | −1.03 | 2.32 | 0.00 |
| **Family stress/conflicts** (Reference group = low level of stress) | | 0.14 | 0.00 | | 0.02 | 0.00 | | 0.078 | 0.00 |
| Moderate level of stress/conflicts | −0.31 | 0.02 | 0.00 | 0.14 | 0.03 | 0.00 | −0.19 | 0.10 | 0.00 |
| High level of stress/conflicts | 1.88 | 0.24 | 0.00 | −0.08 | 0.25 | 0.00 | 0.28 | 0.05 | 0.00 |
| **Academic performance** | −1.46 | 6.86* | 0.04 | −0.42 | 3.71 | 0.00 | −0.32 | 3.19 | 0.00 |
| | $R^2 = 0.026$ | | | $R^2 = 0.018$ | | | $R^2 = 0.023$ | | |

*$p < 0.01$

**$p < 0.001$, B = Unstandardized Regression Coefficient, F = F statistics, $\eta^2$: Eta squared. TRF: Teacher Report Form, YSR: Youth Self Report.

Academic performance was negatively associated with the teacher-adolescent discrepancy score, suggesting that when academic performance increased, TRF scores tended to be lower relative to YSR scores. Further, teacher-adolescent discrepancies were larger for females, and for adolescents from indigenous groups/ethnic minorities.

For Internalizing Problems, 1.8% of the variance in discrepancy scores was explained by the background variables, whereas for Externalizing Problems, 2.3% of the variance was explained by the background variables. For both types of problems, gender and caste/ethnicity were significantly associated with the discrepancy scores: discrepancies were larger for girls and indigenous groups/ethnic minorities. However, the effect sizes were small ($\eta^2 \le 0.01$) for all situations (Table 6).

## Discussion

The present study assessed and compared the mean scale scores for the ninety common items on the YSR, CBCL, and TRF and examined cross-informant correlations and background variables influencing parent-adolescent and teacher-adolescent discrepancies.

### Comparing youth, parent, and teacher reports on mean scores of EBPs

We found that the mean scores of adolescent self-reports were significantly higher than those of parent reports for all problem scales, which is consistent with findings from other international studies [4, 8, 10, 14, 16]. One explanation for this may be that parent reports of EBPs are limited to parents' observations of their child in the home and family settings, and that parents tend to perceive only more overt behaviors [41]. Other researchers have argued that lower emotional problems in parent reports might be suggestive of parental inability to recognize emotional distress and difficulties in their children [21]. Parents may not be aware of their adolescents' emotional problems due to a lack of knowledge about mental health problems or poor parent-child communication. Conversely, adolescents who have insights into their internal states and psychological disturbances, not necessarily accessed by others, are inclined to report them, even if those disturbances are small. This, too, may contribute to differences in EBP scores between parents and adolescents.

The lower mean scores in parent reports could also reflect a reluctance on the part of Nepali adolescents to confide their problems to their parents for fear of being judged. Indeed, this might be the case in LMICs, as mental health problems are often associated with stigma and discrimination in these societies, including in Nepal [42]. Another reason for the lower problems reported by Nepali parents might be related to cultural expectations, parental values, and norms regarding child behavior. Parents in Asian countries such as Nepal value obedience and politeness in their children; hence, they encourage inhibited behaviors like being quiet, avoiding arguments, and being non-aggressive and non-violent [43]. Consequently, they may not perceive inhibited behaviors in their child as a potential indicator of emotional problems. As culture-specific mechanisms might give rise to cross-informant discrepancies [8], future studies in Nepal are warranted and should explore the impact of parental cultural norms, as well as the impact of the stigma of mental health problems on parent-child agreement on EBPs.

Consistent with parent-adolescent reports on EBPs, adolescents reported more problems than their teachers on all problem scales. This finding is consistent with findings from other international studies [7, 26, 27]. Teachers observe adolescents in a structured school environment, where academic performance, attention to tasks, and social interactions with peers and authority figures may be more important than focusing on students' psychological problems. In addition, Nepali teachers' difficulties in recognizing adolescents' psychological distress may be due to a lack of knowledge of mental health problems which might explain some of the teacher-adolescent discrepancies. Another reason might be related to the teacher-student relationship. In Nepal, teachers are perceived as authority figures; hence teacher-student interactions are few and formal, which might limit the teacher's ability to recognize adolescents' emotional distress. Moreover, given that Nepal is home to people from 142 diverse castes/ethnic groups [44], teachers and students may have varied cultural backgrounds, potentially limiting teachers' capacity to identify and report students' mental health problems. Further, in a typical Nepali school, the average teacher-student ratio is 1:28 at the secondary level [45], which makes it difficult for teachers to pay attention to individual students. Consequently, teachers may not be fully aware of the extent of their students' emotional and behavioral difficulties, potentially diminishing their likelihood of reporting such issues.

## Agreement between parents, teachers, and adolescents on EBPs

In the present study, parent-adolescent agreement on Total Problems was moderate ($r = 0.31$). Across the eight syndrome scales, agreement was low to moderate, with correlations ($r$s) ranging from 0.21 to 0.33. These findings corroborate with international studies [4, 8, 12, 14]. However, our findings are in contrast with suggestions from some studies that parent-adolescent discrepancies may be smaller in societies where cultural values promote familism and collectivism, and where parents typically have very close relationships with their adolescents (i.e., in many Asian cultures, including Nepal), than in societies that promote individualism and autonomy (i.e., in many Western cultures) [14, 16]. However, the moderate to low parent-adolescent agreement found in our study suggests that more studies from collectivistic LMICs are warranted to examine cultural variations.

Consistent with previous meta-studies [14, 22], we found a higher parent-adolescent agreement for Externalizing Problems than for Internalizing Problems, which might be due to the higher degree of observability of externalizing behaviors. For the syndrome scales, the parent-adolescent agreement was low for Anxious/Depressed, Withdrawn/Depressed, Attention Problems, and Thought Problems. The weakest correlation was found for the Withdrawn/Depressed scale, possibly because these problems are not readily noticeable. Hence, Nepali parents may have difficulties in recognizing adolescents' internalizing problems. Another

reason behind this might be related to the higher tolerance for internalizing problems and lower tolerance for externalizing problems in Asian cultures [46].

We found a low level of teacher-adolescent agreement for all types of problems, which aligns with findings from Western cultures [7, 25, 27]. The highest teacher-adolescent agreement in our study was found for Externalizing Problems, followed by Attention Problems and Somatic Complaints, whereas the lowest level of agreement was found for Thought Problems. The higher agreement for Externalizing Problems supports the established explanation that teachers are more likely to notice externalizing behaviors because of their overt nature and their likelihood to affect the classroom/school environment [4]. Similarly, Attention Problems are more likely to be recognized by teachers since they interfere with normal classroom functioning and negatively influence students' academic performance [47]. Further, Somatic Problems are often readily expressed by many children and adolescents in Asian cultures, which makes them more easily recognized by teachers. On the other hand, the lower teacher-adolescent agreement on Internalizing Problems may be related to the teachers' inability to identify those kinds of problems, as they are less likely to affect normal classroom functioning [48]. However, there might be other reasons; for instance, some studies have suggested that low cross-informant agreement in population samples may reflect that less stable, less severe, or situation-specific emotional symptoms are difficult to recognize [49].

## Factors associated with cross-informant discrepancies

The present study also explored background variables that may be associated with parent-adolescent and teacher-adolescent discrepancies. Consistent with previous studies on cross-informant discrepancies in EBPs [4, 16, 18], we found that gender, caste/ethnicity, family stress/conflicts, and academic performance affected these discrepancies. However, the effect sizes were small in all situations indicating a minor role of those variables compared to other variables not measured in the study. Small effects can also be a consequence of different behavior in various settings, or different access to information for adolescents, teachers, and parents. It should be mentioned that in the present study, we did not find any impact of factors such as the age of the adolescents and experience of negative/ traumatic life events on cross-informant discrepancies.

## The impact of gender

We found that parent-adolescent discrepancies for the Internalizing Problems and teacher-adolescent discrepancies for Total Problems, Internalizing Problems, and Externalizing Problems were higher for girls than for boys. These findings suggest that there might be problems experienced by adolescent girls that are not perceived by their parents or teachers, particularly emotional problems. The reasons for this may be that emotional problems are not as readily expressed by female adolescents as by male adolescents, or they may be inherently difficult to recognize. Also, the larger discrepancies in the reports for girls could be due to different social expectations regarding behavior in girls versus boys [50]. One explanation could be that both parents and teachers perceive withdrawal or depressive and anxious behaviors as normal in girls and hence overlook these EBPs in girls but not in boys. Further, patterns of parent-adolescent communication and subsequent parental knowledge about adolescents' problem behaviors might differ across genders, leading to a larger discrepancy for girls [51]. As for teacher-adolescent discrepancies in internalizing problems, it has been documented that teachers are likely to favor boys and give them more attention in a typical classroom setting [52]. Hence, the level of teacher engagement across genders might differ and limit the teacher's ability to recognize emotional distress among adolescent girls. Further, girls' tendency to internalize

their difficulties might limit teachers' ability to recognize their problems, which might lead to underreporting. In the present study, we did not explore any possible explanatory factors related to the impact of gender on rating discrepancies. Future studies are warranted to confirm if any of the above explanations apply to Nepal.

## The impact of caste/ethnicity

In the present study, caste/ethnicity was associated both with parent-adolescent and teacher-adolescent discrepancies indicating cross-cultural differences in the pattern of reporting adolescent difficulties by parents and teachers versus the adolescents themselves. Consistent with previous studies [53, 54] which showed a relatively low level of parent-reported and teacher-reported problems compared to that of minority adolescents, the parent-adolescent discrepancies and teacher-adolescent discrepancies for Total Problems, Internalizing Problems, and Externalizing Problems were larger for the indigenous /ethnic minorities group than for the Hindu majority group. However, our findings contrasted with a recent study performed in Australia, which suggested a higher parent-adolescent agreement in indigenous dyads than in non-indigenous dyads [18]. As reported in other studies [55], our minority parents may have a higher threshold for considering child behaviors as abnormal and thus may have been less likely than the parents of ethnic majority children to perceive EBPs [56]. We do not know the reason for the larger parent-adolescent discrepancies in indigenous groups/ethnic minorities in our study. Future studies on indigenous/ethnic minorities in Nepal are warranted to explore further, including the impact of family factors, such as family cohesion, parent-child relationships, and parenting practices.

Further, our findings suggest that the larger teacher-adolescent discrepancies reflect an under-identification of EBPs in ethnic minority groups relative to majority groups, which is consistent with other studies [53]. Teachers may not be primed to expect the adjustment difficulties of minority adolescents; rather they may attribute observed behaviors to racial stereotypes instead of possible EBPs. In Nepal, teachers may be less likely to report problems that are counter to prevailing racial stereotypes, since ethnic stereotypes are quite prevalent [57]. Alternatively, problems may manifest differently among minority adolescents, making these problems more difficult for teachers to recognize [53].

## The impact of family stress/conflicts

Our results indicated that in families with a high level of stress/conflicts, parent-adolescent discrepancies were higher for Total Problems and Internalizing Problems. This finding is consistent with other studies, which have suggested that there are more parent-adolescent discrepancies in problem ratings when adolescents live in an undesirable family environment [16, 58]. This might be because parents are unable to pay enough attention to their children or provide adequate care to prevent possible psychological problems when they experience family conflicts. Alternatively, it might be due to poor communication and the effect of stress influencing the parents' perception of their adolescents' emotions and behaviors [59]. Hence, adolescents from families with a high level of conflict are likely to have more EBPs than their parents can recognize. However, the teacher-adolescent discrepancies were not associated with adolescents' experiences of family stress/conflicts.

## The impact of academic performance

Academic performance was negatively associated with teacher-adolescent discrepancies in Total Problems. For high-performing students, the teachers tended to score the adolescents with fewer mental health problems than the adolescents themselves would, while the opposite

was the case for low-performing students. This finding is in line with other studies that have shown that students who perform better in school tend to have fewer teacher-reported mental health problems [60], but more self-reported problems, leading to larger TRF-YSR discrepancies. On the other hand, teachers are more likely to report higher levels of mental health problems among adolescents with school-specific difficulties [12, 61].

## Strengths and limitations

To our knowledge, the present study is the first school-based study in Nepal that compares adolescents' reports of EBPs with those of their parents and teachers. The study used standardized measures and sound methodology with a random selection of study participants from a large sample of adolescents from several districts in various parts of the country. The participation rate was exceptionally high (99%), probably due to a thorough and closely monitored data collection procedure.

There are some inherent limitations in the study, and findings should be interpreted accordingly. Firstly, we cannot claim that the results are representative of the whole of Nepal due to the purposive selection of districts and schools and the fact that not all districts in the country were included. The use of probability sampling / random selection of districts and schools would have been more robust. Secondly, our study was a school-based study that did not include adolescents who were not present at school at the time of data collection. Because of this, our results might be biased as the group of adolescents who did not attend school might have had a higher level of EBPs / mental disorders [62]. Third, we did not include a separate clinical sample for comparison. Generally, the cross-informant agreement is higher in referred samples [63, 64]. In future Nepali studies, an examination of cross-informant agreement of EBPs in adolescents referred to child and adolescent services is recommended to get an estimate of cross-informant agreement on EBPs in clinical samples. Further, few adolescents from the low caste group and the indigenous/ethnic minority group were included in the study as the sample was not stratified for caste/ethnicity during the selection of participants. Because of the small numbers in certain ethnic groups, a cross-cultural comparison of the cross-informant agreement was not performed. If included, this might have resulted in more accurate estimates of EBPs. The present study used quantitative methods that only rated adolescent behavior. Additional qualitative methods, such as observational methods and interviews, could have provided a deeper understanding of the phenomena.

Further, our study involved a somewhat vague component related to negative/traumatic life events. Using a single question to assess negative/traumatic life events and not a validated questionnaire may have limited the validity of the measurement of the traumatic life events. A general item may not capture the complexity of trauma as experienced by individuals, which affects the ability to detect associations. Additionally, there might be factors other than those examined in this study that might have affected cross-informant discrepancies, such as maternal psychopathology [65, 66] or the mother-child relationship [67]. Similarly, adolescents' social functioning, and school engagement, might have affected teacher-adolescent discrepancies [12]. Incorporating measures of school engagement, such as school attendance or changes in academic performance, could have helped to elucidate aspects of the school environment that may be associated with cross-informant discrepancies.

## Clinical implications

The most common outcome of multi-informant assessments of adolescents' EBPs is discrepancies across informants' reports. These discrepancies often provide domain-specific information and demonstrate variations in informants' perspectives and expertise for observing

adolescents within specific contexts. Hence, collecting and interpreting multi-informant data comprise "best practices" in clinical care [6]. Cross-informant discrepancies in adolescent EBPs as were demonstrated in the present study, might have important implications for treatment. Studies have shown that cross-informant discrepancies are linked to poorer treatment outcomes in adolescents [68, 69] and that fewer discrepancies between caregivers and adolescents predict improved treatment engagement and outcomes [70, 71]. Low cross-informant agreement can undermine treatment planning and implementation [6], posing difficulties in making a valid diagnosis and subsequent treatment decisions, ultimately resulting in poor treatment compliance.

The moderate to low agreement found in our study underscores the importance of multi-informant assessment in clinical practice to obtain a holistic view of the adolescent's functioning across different contexts. Low informant agreement between informants who observe an adolescent in unique contexts (i.e., home, school) should lead to a careful consideration of context-specific manifestations of mental health problems. Discrepancies in reports highlight the need for interventions that are sensitive to context-specific behaviors and may suggest different strategies for home versus school environments. Furthermore, poor agreement on symptoms may reflect communication difficulties, highlighting the need for improved communication and collaboration between mental health professionals, teachers, parents, and adolescents to identify symptoms accurately occurring in different environments.

Finally, our study suggests that clinicians need to pay particular attention to adolescents' reports of internalizing problems which may be more likely to be reported by adolescents than other informants. The study also emphasizes the importance of considering the impact of contextual and cultural factors on cross-informant discrepancies.

## Future research

In future empirical studies, we would recommend focusing more on cross-informant discrepancies in different castes and ethnic groups in Nepal. Further, other factors not examined in the present study that might impact cross-informant discrepancies, such as parental psychopathology, parent-child attachment, and school engagement, are recommended as future research topics. Such studies might provide a more comprehensive picture of the mechanisms involved. Finally, studies linking informant discrepancies to service outcomes, and clinical studies focusing on strategies for integrating multi-informant data in adolescent mental health treatments are warranted for future research.

## Conclusion

The present study is the first large-scale study from Nepal that compares adolescents' reports of EBPs with those of their parents and teachers. It sheds light on the cross-informant agreement and the factors associated with cross-informant discrepancies, providing valuable insights for clinicians, researchers, and policymakers in their effort to develop child and adolescent mental health services in the country. Consistent with international studies, adult informants (parents and teachers) reported fewer problems than adolescents themselves, and the parent-adolescent and teacher-adolescent agreements were moderate to low. Gender, caste/ethnicity, family stress/conflicts, and academic performance were associated with cross-informant discrepancies which need to be taken into consideration when assessing EBPs in Nepali adolescents. Further studies focusing on the cross-informant agreement and factors impacting cross-informant discrepancies might highlight the importance of integrating multiple informants' reports in the assessment of adolescents' mental health problems.

## Supporting information

**S1 Dataset. YSR, CBCL, and TRF reports for Nepali school-going adolescents.**
(XLSX)

## Acknowledgments

We are grateful to all participating Nepali adolescents, their parents and teachers, and the team of data enumerators and supervisors for making this study possible. Further, we would like to extend our gratitude to the child and adolescent psychiatry team at Kanti Children's Hospital, Kathmandu, for their support.

## Author Contributions

**Conceptualization:** Sirjana Adhikari, Per Håkan Brøndbo, Anne Cecilie Javo.

**Data curation:** Sirjana Adhikari, Jasmine Ma, Bjørn Helge Handegård.

**Formal analysis:** Sirjana Adhikari, Bjørn Helge Handegård.

**Funding acquisition:** Anne Cecilie Javo.

**Methodology:** Jasmine Ma, Bjørn Helge Handegård, Anne Cecilie Javo.

**Project administration:** Jasmine Ma.

**Supervision:** Suraj Shakya, Per Håkan Brøndbo, Bjørn Helge Handegård, Anne Cecilie Javo.

**Writing – original draft:** Sirjana Adhikari.

**Writing – review & editing:** Sirjana Adhikari, Jasmine Ma, Suraj Shakya, Per Håkan Brøndbo, Bjørn Helge Handegård, Anne Cecilie Javo.

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
