## [Decision Letter · Decision Letter 0]

26 Jan 2024

PONE-D-23-34622Cross-informant correlations on emotional and behavioral problems in Nepali adolescents. Do problems reported by adolescents differ from those reported by their parents and teachers?PLOS ONE

Dear Dr. Adhikari,

Thank you for submitting your manuscript to PLOS ONE. After careful consideration, we feel that it has merit but does not fully meet PLOS ONE’s publication criteria as it currently stands. Therefore, we invite you to submit a revised version of the manuscript that addresses the points raised during the review process.

We look forward to receiving your revised manuscript.

Kind regards,

Laura Kelly

Division Editor

PLOS ONE

Journal Requirements:

"Sirjana Adhikari (SA) received funding for the study from the NORPART 2018/10039 project (“Collaboration in Higher Education in Mental Health between Nepal and Norway”) and Child Workers In Nepal (CWIN-Nepal). "

3. We are unable to open your Supporting Information file [Supporting information (S1)_Sirjana Adhikari.sav]. Please kindly revise as necessary and re-upload.

**Additional Editor Comments:**

The reviewers have raised a number of concerns that need attention. They request additional information on the emotional and behavioral problem indicators as well as some further analysis, including a comparison of assessments between parents and teachers. Please also expand on research limitations. 

Reviewers' comments:

Reviewer's Responses to Questions

**Comments to the Author**

1. Is the manuscript technically sound, and do the data support the conclusions?

Reviewer #1: Yes

Reviewer #2: Partly

2. Has the statistical analysis been performed appropriately and rigorously? 

Reviewer #1: Yes

Reviewer #2: Yes

3. Have the authors made all data underlying the findings in their manuscript fully available?

Reviewer #1: Yes

Reviewer #2: Yes

4. Is the manuscript presented in an intelligible fashion and written in standard English?

Reviewer #1: Yes

Reviewer #2: Yes

5. Review Comments to the Author

Reviewer #1: 1. In the abstract, results, and conclusion, similarities can be acknowledged. The author may include recommendations in the conclusion.

2. The introduction section of the full paper lacks a statement of the problem for the current study.

3. In line 136, the third aim employs the word "exploring," which appears to be more qualitative in nature. This word should be reframed.

4. Cronbach's alpha should be mentioned in the Youth Self Report, Child Behavior Checklist, and Teacher’s Report Form.

5. Please use a more appropriate term in line 481; instead of "mental symptoms," it should be reframed.

6. Strengths and recommendations should be emphasized more in terms of further research scope in the conclusion section.

7. Negative results can also be included in the discussion section.

Reviewer #2: The manuscript delves into a pertinent theme and is well articulated. However, it falls short of being entirely groundbreaking, especially given its focus on a specific population of Nepalese adolescents. This aspect wasn't adequately addressed in the data discussion. Moreover, the clinical implications of the observed low agreement among adolescents, their parents, and teachers concerning emotional and behavioral problems require further exploration.

The statistical analyses conducted are appropriate and align with the study's objectives, albeit being relatively straightforward (t-test, correlation, and regression). I suggest including descriptive analyses of emotional and behavioral problem indicators to comprehend the percentage of the sample classified as normal, borderline, or clinical. Additionally, a comparison of assessments between parents and teachers regarding adolescents' emotional and behavioral problems is recommended. Teachers, with their close interaction and potential to observe changes, are crucial in mental health referrals. Lastly, it's noteworthy that the question about serious life events/trauma used in the regression analysis was formulated by the authors and is not part of a validated questionnaire, potentially weakening the analysis.

The manuscript is written in clear and cohesive language. However, I recommend organizing the text more integratively, emphasizing the interrelation among the various aspects considered in the study. This would facilitate a more comprehensive discussion.

Out of the 65 references cited, only 23 have been published in the last five years. Furthermore, several references lack complete data, such as DOIs. Providing this information is crucial to facilitating readers' access to the cited documents. I recommend ensuring that all references, including DOIs, are complete to enhance the comprehensiveness and accessibility of the bibliography.

6. PLOS authors have the option to publish the peer review history of their article (what does this mean?). If published, this will include your full peer review and any attached files.

Reviewer #1: No

Reviewer #2: No

---

## [Author Response · Author response to Decision Letter 0]

9 Mar 2024

Response to the Reviewers’ comments

Dear Reviewers,

We would like to thank you for reviewing our manuscript. Your comments and questions have been valuable in improving the quality of our paper. We have revised the manuscript according to your advice, to the best of our knowledge. References have been revised accordingly. We have also done additional proofreading, correcting some words and sentences, but without changing the meanings and not adding any more information to the text. 

Below, we have given our answers to your comments one by one. In doing so, we have referred to the corresponding lines in our revised manuscript with track changes (see Revised manuscript with track changes). 

Reviewer #1

Reviewer’s comment

1. In the abstract, results, and conclusion, similarities can be acknowledged. The author may include recommendations in the conclusion.

Our Response: 

Thank you for the suggestion. We have now revised the conclusion part of the abstract, including implications for clinical practice – see lines 46-52. 

Reviewer’s comment

2. The introduction section of the full paper lacks a statement of the problem for the current study.

Our Response: 

Thank you for the comment. We have now included the statement of problems for the current study, see the last paragraph of the introduction, lines 131 – 132, 138-145. 

Reviewer’s comment

3. In line 136, the third aim employs the word "exploring," which appears to be more qualitative. This word should be reframed.

Our Response

 Thank you for the suggestion. We agree and have now changed the word “exploring” to “examining”, see line 149. 

Reviewer’s comment

4. Cronbach's alpha should be mentioned in the Youth Self Report, Child Behavior Checklist, and Teacher’s Report Form.

Our Response: 

Thank you for the suggestion. We have added Cronbach’s alpha for the syndrome scales of the Child Behavior Checklist/6-18 years, and Teacher’s Report Form/6-18 years, in lines 217-221. In addition, for the reader's convenience and to provide a comprehensive overview of the psychometric properties of the syndrome scales across different measures, we have included a compiled table of Cronbach's alpha values, see lines 230-235. 

Reviewer’s comment

5. Please use a more appropriate term in line 481; instead of "mental symptoms," it should be reframed.

Our Response: 

Thank you for the comment. We have now used “mental health problems” instead of “mental symptoms” in the corresponding line (see line 597). 

Reviewer’s comment

6. Strengths and recommendations should be emphasized more in terms of further research scope in the conclusion section.

Our Response: 

Thank you for the comment. We have rearranged and extended the limitation part in the “ Strengths and Limitations” paragraph, see lines 606-663. Further, we have moved the recommendations for further research from the Strengths and Limitations part to a new paragraph called “Future research” to emphasize our recommendations, see lines 660-667. 

Reviewer’s comment

7. Negative results can also be included in the discussion section.

Our Response:

Thank you for the comment. We have now included negative findings in the discussion section in lines 530-532 “It should be mentioned that in the present study, we did not find any impact other factors, such as the age of the adolescents and the experience of negative/traumatic life events on cross-informant discrepancies”, and also in lines 588-589 as “However, the teacher-adolescent discrepancies were not associated with adolescents’ experience of family stress/conflicts.” 

Reviewer #2

Reviewer’s comment

The manuscript delves into a pertinent theme and is well articulated. However, it falls short of being entirely groundbreaking, especially given its focus on a specific population of Nepalese adolescents. This aspect wasn't adequately addressed in the data discussion. Moreover, the clinical implications of the observed low agreement among adolescents, their parents, and teachers concerning emotional and behavioral problems require further exploration.

Our Response: 

Thank you for your comments. In the Strengths and Limitations part of the Discussion section, we have argued that the study is the first large-scale study of its kind in Nepal, that it was conducted in many districts in different parts of the country, and that the participation rate was exceptionally high, in lines 600-605. However, we have commented more as to the limitations of the study sample, see lines 606-633. Further, we have commented more on the clinical implications of the low cross-informant agreement in the Clinical Implications part of the Discussion section, see lines 634-659. 

Reviewer’s comment

The statistical analyses conducted are appropriate and align with the study's objectives, albeit being relatively straightforward (t-test, correlation, and regression). I suggest including descriptive analyses of emotional and behavioral problem indicators to comprehend the percentage of the sample classified as normal, borderline, or clinical. Additionally, a comparison of assessments between parents and teachers regarding adolescents' emotional and behavioral problems is recommended. Teachers, with their close interaction and potential to observe changes, are crucial in mental health referrals. 

Our Response:

Thank you for the comments. We agree that examining the prevalences of adolescent EBPs by presenting data from the YSR, CBCL, and TRF might be of interest. However, we chose to compare the mean scores (magnitude of problems) only, as Nepali norms do not yet exist, and the estimates would be based on available American norms. As the validity of our findings when using the American norms might be questionable as to relevant cut-off points, we decided to use the raw scores (mean scores) that are independent of cultural norms. Hopefully, future studies may provide appropriate norms for Nepali children and adolescents. 

Another thing is that if we extend the study to include prevalence tables and comments/discussion about those findings as well, it might make the study less focused on its main topic, i.e., the cross-informant correlations. We would like to inform you that in our preceding paper on self-reported EBPs among school-going adolescents in Nepal published in 2023 (same sample), we already presented adolescent-reported prevalences and percentages classified as normal, borderline, and clinical, and for all problem scales (Adhikari S, Ma J, Shakya S, Brøndbo PH, Handegård BH, Javo AC. Self-reported emotional and behavioral problems among school-going adolescents in Nepal - A cross-sectional study. PLOS ONE. 2023;18(6):e0287305. doi: 10.1371/journal.pone.0287305. PMID: 37352299). The reason for not including a parent-teacher comparison of adolescent EBPs is that we wanted to avoid replication. A previous paper by Ma and colleagues had already examined parents’ and teacher’s reports for the wider age group of children and adolescents (6-18 years), which also included the present study sample of adolescents (Ma, J., Mahat, P., Brøndbo, P.H. et al.: Teacher reports of emotional and behavioral problems in Nepali schoolchildren: to what extent do they agree with parent reports? BMC Psychiatry 22, 584 (2022). https://doi.org/10.1186/s12888-022-04215-4). 

Reviewer’s comment

Lastly, it's noteworthy that the question about serious life events/trauma used in the regression analysis was formulated by the authors and is not part of a validated questionnaire, potentially weakening the analysis.

Our Response: 

Thank you for rightly pointing out that limitation. We now mentioned that in the limitations section, lines 623-627. 

Reviewer’s comment

The manuscript is written in clear and cohesive language. However, I recommend organizing the text more integratively, emphasizing the interrelation among the various aspects considered in the study. This would facilitate a more comprehensive discussion.

Our Response:

Thank you for your constructive feedback. We have now reorganized the Discussion section to improve text integration. The first paragraph now includes the comparison of mean scores of parents, teachers, and the adolescents themselves. The next paragraph focuses on correlations. Instead of two separate discussions under separate headings, one about the parent-adolescent agreement and one about the teacher-adolescent agreement, we have now merged the discussions under one common heading (see lines 369-446). 

Reviewer’s comment

Out of the 65 references cited, only 23 have been published in the last five years. Furthermore, several references lack complete data, such as DOIs. Providing this information is crucial to facilitating readers' access to the cited documents. I recommend ensuring that all references, including DOIs, are complete to enhance the comprehensiveness and accessibility of the bibliography.

Our Response:

Thank you for the comment. We have now completed the bibliography section with all required data, such as DOI, wherever applicable. We also have added a few new references in the text (lines 67, 104, 404, 439, 612) and the references (lines 719-726, 785-789, 856-859, 868-870, 920-922.

---

## [Decision Letter · Decision Letter 1]

3 Apr 2024

PONE-D-23-34622R1Cross-informant correlations on emotional and behavioral problems in Nepali adolescents. Do problems reported by adolescents differ from those reported by their parents and teachers?PLOS ONE

Dear Dr. Adhikari,

Thank you for submitting your manuscript to PLOS ONE. After careful consideration, we feel that it has merit but does not fully meet PLOS ONE’s publication criteria as it currently stands. Therefore, we invite you to submit a revised version of the manuscript that addresses the points raised during the review process.

We look forward to receiving your revised manuscript.

Kind regards,

Harikrishnan U, Ph.D

Guest Editor

PLOS ONE

Journal Requirements:

Reviewers' comments:

Reviewer's Responses to Questions

**Comments to the Author**

1. If the authors have adequately addressed your comments raised in a previous round of review and you feel that this manuscript is now acceptable for publication, you may indicate that here to bypass the “Comments to the Author” section, enter your conflict of interest statement in the “Confidential to Editor” section, and submit your "Accept" recommendation.

Reviewer #3: All comments have been addressed

2. Is the manuscript technically sound, and do the data support the conclusions?

Reviewer #3: Yes

3. Has the statistical analysis been performed appropriately and rigorously? 

Reviewer #3: Yes

4. Have the authors made all data underlying the findings in their manuscript fully available?

Reviewer #3: Yes

5. Is the manuscript presented in an intelligible fashion and written in standard English?

Reviewer #3: Yes

6. Review Comments to the Author

Reviewer #3: Title Modification is required

Scientific Rationale for Selecting Six Students from Each Grade

Explanation of Statistical Analysis: Utilizing the intraclass correlation coefficient (ICC) instead of Pearson's correlation coefficient can offer several advantages, particularly in research contexts involving multiple raters or repeated measurements.

7. PLOS authors have the option to publish the peer review history of their article (what does this mean?). If published, this will include your full peer review and any attached files.

Reviewer #3: No

---

## [Author Response · Author response to Decision Letter 1]

26 Apr 2024

Response to Reviewer

Reviewer’s comment 

 Overall, it is a well-conducted study. 

Our response 

 Thank you for the positive comment.

. 

Reviewer’s comment Cross-informant correlations on emotional and behavioral problems in Nepali adolescents. Do problems reported by adolescents differ from those reported by their parents and teachers? 

 The title requires modification. 

Our response Thank you for the comment. We have changed the title to a more appropriate one now.: “Cross-informant ratings on emotional and behavioral problems in Nepali adolescents: comparison of adolescents’ self-reports with parents’ and teachers’ reports”. See lines 1-3 in the manuscript. 

Reviewer’s comment 

 Based on the population distribution of 

the three main ecological/geographical regions of Nepal, sixteen districts were selected (three from 

the Mountains region, six from the Middle Hills region, six from the Terai region, and the 

Kathmandu district) Any sampling procedure used for the selection of district. 

 In the methodology section, the sampling procedure appears to be weak; utilizing multistage sampling or probability proportionate to size sampling would have been more robust. 

Our response Thank you for pointing out possible weaknesses of the sampling procedure used to select districts. We selected districts as we selected schools using a purposive sampling method based on accessibility and feasibility. We have now clarified it in the manuscript and explained the reasons for choosing this method in lines 157-160. We acknowledge that the random selection of districts and schools using probability sampling would have been more robust. However, this was a large, countrywide study that required an extensive amount of time and money to accomplish. Therefore, we chose the purposive sampling technique for cost-effectiveness and ease of data collection and travel. Nevertheless, we randomly selected the students from each grade using simple random sampling. As we recognize that purposive sampling to select districts and schools was a limitation of our study, we have also added a few more lines about it in the limitation section, lines 520-522.

Reviewer’s comment 

 Six adolescents 

(three boys and three girls aged 11-18 years) from each grade level (grades 6-10) were randomly 

selected using random number tables, regardless of caste/ethnicity or other background variables. 

The overall participation rate was 99.2% for the adolescent sample and parent sample, and 99.1% 

for the teacher sample. What scientific rationale supports the selection of only six students from each grade? 

Our response We selected six students only from each grade for a couple of reasons. One reason was that some government schools in Nepal, mostly in rural areas, have very few students (less than 10 students per class), whereas private schools usually have more students per class. To ensure an equal number of students from both government and private schools, we decided on a smaller number of students from each grade. Another reason was to keep the burden of filling out the TRF reasonable for teachers, who would have to answer more than 150 questions per student. An additional reason to keep the number of students per classroom low was our concern about the dependency of scores from clustering of students within classrooms when the same teacher informs about multiple students. Sampling many students within a classroom increases the design effect, and thus reduces the potential benefit of increased within-classroom sample size.

Reviewer’s comment 

 Statistical analysis The researcher opted for Pearson's correlation but employing the intraclass correlation coefficient (ICC) would have been more advantageous. COMMENTS 

Using the intraclass correlation coefficient (ICC) instead of Pearson's correlation coefficient can indeed be more appropriate in certain research contexts, particularly when dealing with data that involve multiple raters or measurements on the same subjects.

Pearson's correlation coefficient measures the strength and direction of the linear relationship between two variables. It assumes independence of observations and is primarily used for assessing the relationship between two continuous variables.

On the other hand, the ICC is a statistical measure used to assess the reliability and consistency of measurements made by multiple observers or raters, or by the same observer or rater on different occasions. It is particularly useful in situations where there is a need to quantify the agreement or reliability among different raters or between different measurements made on the same subjects.

In research settings where there are multiple raters providing ratings or measurements on the same set of subjects, ICC provides a more robust measure of reliability compared to Pearson's correlation coefficient. It takes into account both systematic and random variations in the data and provides a measure of consistency or agreement among the raters or measurements.

Therefore, if the researcher is dealing with data that involve multiple raters or repeated measurements, using ICC would likely provide a more appropriate measure of reliability compared to Pearson's correlation coefficient. However, the choice between these two measures ultimately depends on the specific research question and the nature of the data being analyzed.

Our response Thank you for the comment. We acknowledge that using the intraclass correlation coefficient (ICC) instead of Pearson's correlation coefficient would have been more advantageous if the focus of the study was on assessing the reliability of the measurements. In a situation where the raters view the adolescent from the same angle (for example in an observational situation or from a video recording) using the same set of items, it would be more natural to report reliability. However, in the specific context of our study which primarily focused on examining the agreement among different informants, such as teachers, parents, and adolescents, who view the child’s behavior in different settings and from different “angles”, we believe that reporting Pearson’s correlation remains the more appropriate choice. Furthermore, the study employed three different measures, namely the Youth Self-Report (YSR), Child Behavior Checklist (CBCL), and Teacher's Report Form (TRF), each with distinct sets of items tailored to capture adolescents' emotional and behavioral problems from different perspectives. Although ICC does not explicitly assume homogeneity of items within different measures, the interpretation can be influenced when the items are markedly different across measures, such as in the YSR, CBCL, and TRF. Therefore, we chose an alternative measure of agreement, such as Pearson's correlation coefficient, which we believe is more appropriate for our study.

---

## [Editor Report · Decision Letter 2]

30 Apr 2024

Cross-informant ratings on emotional and behavioral problems in Nepali Adolescents: a comparison of adolescents’ self-reports with parents’ and teachers’ reports

PONE-D-23-34622R2

Dear Dr. Adhikari,

We’re pleased to inform you that your manuscript has been judged scientifically suitable for publication and will be formally accepted for publication once it meets all outstanding technical requirements.

Kind regards,

Harikrishnan U, Ph.D

Guest Editor

PLOS ONE